# Molecular Phylogenetics of the Orchid Genus *Spathoglottis* (Orchidaceae: Collabieae) in Peninsular Malaysia and Borneo

**Farah Alia Nordin** [1,*], **Kartini Saibeh** [2], **Rusea Go** [3], **Khairul Nasirudin Abu Mangsor** [4] and **Ahmad Sofiman Othman** [1]

1 School of Biological Sciences, Universiti Sains Malaysia, Gelugor 11800, Penang, Malaysia
2 Faculty of Sustainable Agriculture, Universiti Malaysia Sabah, Sandakan 90509, Sabah, Malaysia
3 Department of Biology, Faculty of Science, Universiti Putra Malaysia, Serdang 43400, Selangor, Malaysia
4 Analytical Biochemistry Research Centre, Universiti Sains Malaysia, Gelugor 11800, Penang, Malaysia
* Correspondence: farahalianordin@usm.my; Tel.: +60-46536161

**Abstract:** Phylogenetic relationships of the orchid genus *Spathoglottis* (Orchidaceae: Collabieae) in Peninsular Malaysia and Borneo were inferred using the internal transcribed spacer of a nuclear ribosomal DNA (nrITS), a plastid gene *maturase*K (*mat*K) and the plastid region *trn*L-F. Eleven species and three infraspecific taxa of *Spathoglottis* were examined, with two outgroup species, were included in the phylogenetic analysis. The combined plastid and nuclear data revealed *Spathoglottis* as monophyletic. From the maximum likelihood, maximum parsimony and Bayesian analyses, *Spathoglottis* is divided into four major groups which are, (1) the Dwarf Purple *Spathoglottis*, (2) the Dwarf Yellow *Spathoglottis*, (3) the Large Purple *Spathoglottis*, and (4) the Large Yellow *Spathoglottis*. The split in the Dwarf and Large *Spathoglottis* groups might reflect an early differentiation of plant size, flower colours and flower size. Phylogeny reconstruction of the orchid genus *Spathoglottis* also exhibited strong support towards the taxonomic delimitation of the two mostly debated taxa in the genus, *S. aurea* and *S. microchilina*.

**Keywords:** Orchidaceae; Collabieae; *Spathoglottis*; phylogenetics; Bayesian analysis

## 1. Introduction

The genus *Spathoglottis* Blume (subfamily Epidendroideae, tribe Collabieae) is a well-known genus with a total of 49 terrestrial geophyte species and is widely distributed in tropical and subtropical Asia and the Pacific Islands, where 44 species were recorded in the Malesian region and are concentrated particularly in New Guinea [1]. Eight species and three infraspecific taxa of *Spathoglottis* are recognized as native to Peninsular Malaysia and Borneo, namely *Spathoglottis affinis* de Vriese, *S. aurea* Lindl., *S. confusa* J.J.Sm., *S. gracilis* Rolfe ex Hook.f., *S. hardingiana* C.S.P.Parish & Rchb.f., *S. kimballiana* Hook.f., *S. kimballiana* var. *angustifolia* Ames, *S. kimballiana* var. *kimballiana*, *S. microchilina* Kraenzl., *S. plicata* Blume and *S. plicata* var. *alba*.

Despite being popular in horticulture, *Spathoglottis* is a taxonomically confused genus in Orchidaceae [2–5]. Early revision work on this genus was first discussed for the Australian *Spathoglottis*, later followed by revisions for the Pacific Islands and the New Caledonian species [2,6,7]. However, a comprehensive revision on the genus has been lacking until now, and no molecular phylogenetic examination has ever been attempted before. Investigations on the morphology and anatomy of *Spathoglottis* have been reported by several authors [8–13]. On the cytogenetics part, dysploidy was observed in *Spathoglottis* (chromosome count of 2n = 18, 38, 40, 42, and 60) [14,15]. Updates on the palynology of *Spathoglottis* are very scarce, although pollen characters were suggested to be taxonomically informative.

Recently, a study on the chloroplast DNA barcoding of *Spathoglottis* in Malaysia for genetic conservation has been carried out using four chloroplast regions. The chloroplast



regions which are *mat*K, *rbc*L-a, *rpo*B and *rpo*C1 were successfully amplified from four species and one infraspecies namely *S. aurea*, *S. gracilis*, *S. kimballiana*, *S. plicata* and *S. plicata* var. *alba* [16]. However, the findings do not provide a deep phylogenetic inference, as the main goal was not to determine patterns of relationship but to identify an unknown sample in terms of a preexisting *Spathoglottis* classification.

Among others, the most frequently misidentified species within *Spathoglottis* are the two yellow-flower species with a distinctive narrow lip, *S. aurea* and *S. microchilina.* Over the years, taxonomic status between these two species has been repeatedly questioned. *Spathoglottis aurea* and *S. microchilina* are among the most debated taxa, with checkered histories of species identification [9,17–20]. Morphologically, the two species are distinguished based on the width of the lip (1.5 mm in *S. microchilina*; 4 mm wide in *S. aurea*) and the ability of the flower of *S. microchilina* to self-pollinate (cleistogamy). In the wild, populations of both *S. aurea* and *S. microchilina* show great phenotypic variations and plasticity. Thus, further examination is required to delimit *S. aurea* and *S. microchilina* as two distinct species or just one highly variable form of *S. aurea*.

On a biogeographical note, it is very informative to look at the distribution patterns among the species of this genus. Certain species of *Spathoglottis* are observed to confine to only particular biogeographical regions, with almost a complete no crossing-over between the area of their occurrences, except for the most widespread and weedy species, *S. plicata*. Vicariance hypothesis is assumed for the dispersal pattern of *Spathoglottis* and intercontinental long-dispersal is relatively uncommon or impossible.

It is a pressing need, since establishing species relationships will address issues such as evolutionary relationships between species as well as determination of identity or taxonomic status of certain species. Thus, the present work aimed to elucidate species relationships between members in genus *Spathoglottis* from Peninsular Malaysia and Borneo; and to resolve the taxonomic questions between the controversial yellow-flower *S. aurea* and *S. microchilina*.

## 2. Materials and Methods

### 2.1. Taxon Sampling

A total of 32 accessions from 11 species and three infraspesific taxa of *Spathoglottis* were analyzed in this present work (Figure 1). Two species from tribe Collabieae namely *Tainia paucifolia* (Breda) J.J.Sm. and *Calanthe tankervilleae* (Banks) M.W.Chase, Christenh. & Schuit. were selected as the outgroups. The fresh DNA samples were obtained from different localities in Peninsular Malaysia and Borneo (Sabah and Sarawak). Four *Spathoglottis* species from Thailand, Irian Jaya and New Caledonia; which some of the plant materials were provided by our collaborators were also included in this phylogenetic analysis to examine discreet evolutionary relationships between members of the genus.

The localities for all *Spathoglottis* species and the outgroups used, with voucher information and GenBank accession numbers, are listed in Table 1. All voucher specimens were deposited in the Herbarium, School of Biological Sciences, Universiti Sains Malaysia, Penang, Malaysia (USMP) [21].

### 2.2. Morphological Observations

Morphological examinations were conducted on all 11 species and three infraspesific taxa of *Spathoglottis* analyzed in this present work. Taxa with the plant size below 30 cm tall was categorized as Dwarf *Spathoglottis*, while taxa with the plant size above 30 cm tall; and up to 2 m tall was categorized as Large *Spathoglottis*.

The flower was categorized as small with 3.0 cm cross, medium–sized with 3.0–4.0 cm cross, and large–sized if the flower is >5.0 cm cross when measured while opening. The taxa are also further distinguished based on the shape of the labellum, as either with narrow lip or broad/bilobulate lip.

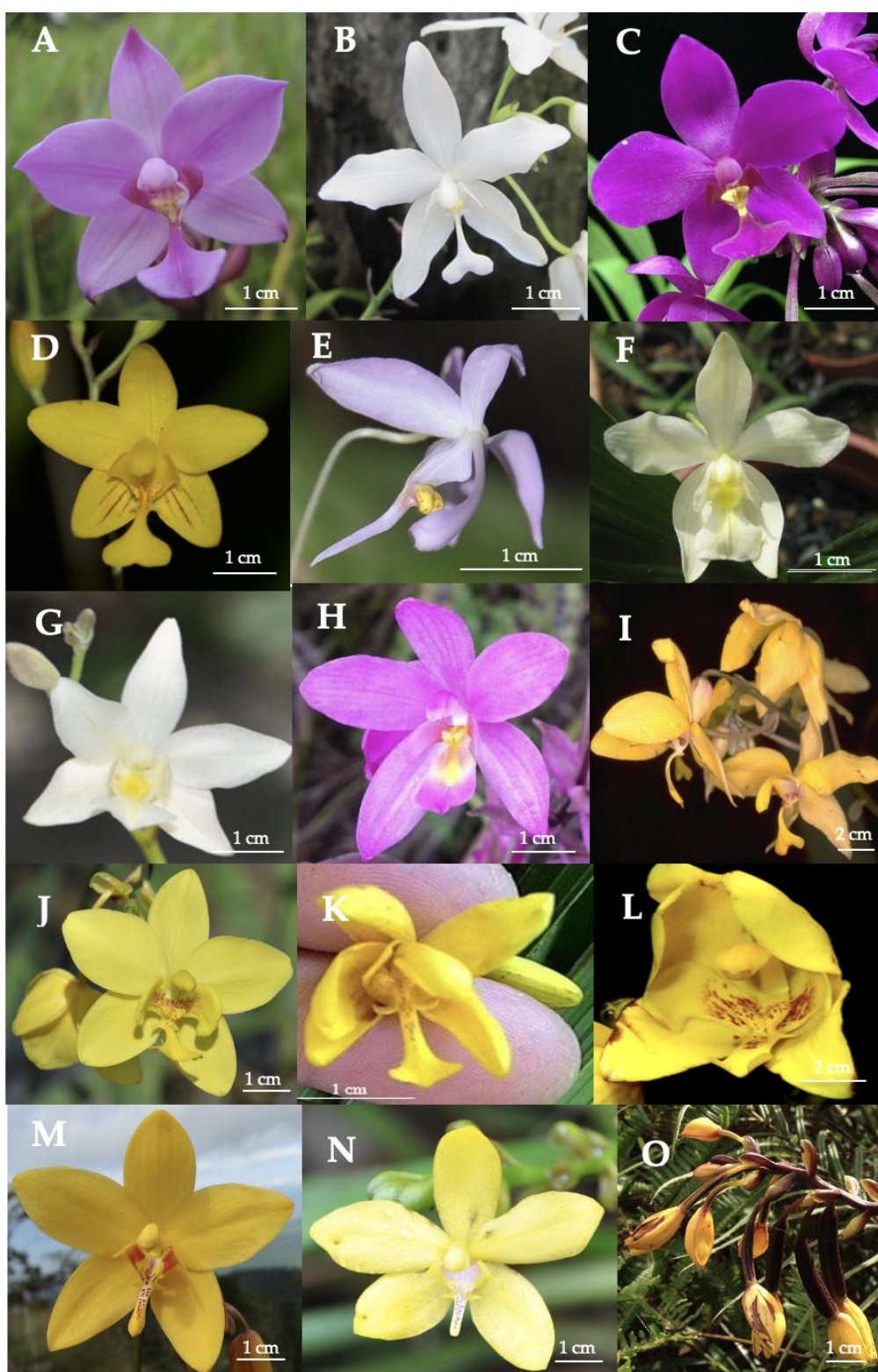

**Figure 1.** The species and infraspecific taxa of *Spathoglottis* from Peninsular Malaysia, Borneo, Thailand, Irian Jaya and the New Caledonia used in this present work: (**A**) *S. plicata* Blume; (**B**) *S. plicata* var. *alba*; (**C**) *S. unguiculata* (Labill.) Rchb.f.; (**D**) *S. affinis* de Vriese; (**E**) *S. hardingiana* C.S.P.Parish & Rchb.f.; (**F**) *S. pubescens* Lindl.; (**G**) *S. eburnea* Gagnep.; (**H**) *S. parviflora* Kraenzl.; (**I**) *S. gracilis* Rolfe ex Hook.f.; (**J**) *S. kimballiana* Hook.f.; (**K**) *S. kimballiana* var. *kimballiana*; (**L**) *S. kimballiana* var. *angustifolia* Ames; (**M**) *S. aurea* Lindl.; (**N**) *S. microchilina* Kraenzl.; and (**O**) cleistogamous flowers of *S. aurea*. Photos: Farah Alia Nordin and Peter O' Byrne (I, K and L: KIP1266f and THH13-6-99).

**Table 1.** Species list, herbarium voucher numbers, localities and GenBank accession numbers of the *Spathoglottis* species and outgroups used in this present work.

| | Species | Voucher Number | Locality | GenBank Accession Number | | |
|---|---|---|---|---|---|---|
| | | | | ITS | *mat*K | *trn*L-F |
| 1. | *Spathoglottis affinis* de Vriese | FAN023 | Fang District, Chiang Mai, Thailand | MG868982 | MG869016 | MG869050 |
| 2. | *Spathoglottis affinis* de Vriese | FAN025 | Thailand-Myanmar Border | MG869002 | MG869036 | MG869070 |
| 3. | *Spathoglottis affinis* de Vriese | FAN028 | Gunung Jerai, Kedah, Malaysia | MG868983 | MG869017 | MG869051 |
| 4. | *Spathoglottis aurea* Lindl. | FAN030 | Fraser's Hill, Pahang, Malaysia | MG868984 | MG869018 | MG869052 |
| 5. | *Spathoglottis aurea* Lindl. | FAN039 | Gunung Ulu Kali, Pahang, Malaysia | MG868985 | MG869019 | MG869053 |
| 6. | *Spathoglottis aurea* Lindl. | FAN040 | Gunung Chin Chin, Pahang, Malaysia | MG868986 | MG869020 | MG869054 |
| 7. | *Spathoglottis aurea* Lindl. | FAN044 | Gunung Jerai, Kedah, Malaysia | MG868987 | MG869021 | MG869055 |
| 8. | *Spathoglottis aurea* Lindl. | FAN052 | Gunung Ulu Kali, Pahang, Malaysia | MG868988 | MG869022 | MG869056 |
| 9. | *Spathoglottis aurea* Lindl. | FAN057 | Gunung Bunga Buah, Selangor, Malaysia | MG868989 | MG869023 | MG869057 |
| 10. | *Spathoglottis aurea* Lindl. | FAN099 | Tanah Rata, Cameron Highlands, Pahang, Malaysia | MG868990 | MG869024 | MG869058 |
| 11. | *Spathoglottis eburnea* Gagnep. | FAN022 | Fang District, Chiang Mai, Thailand | MG868991 | MG869025 | MG869059 |
| 12. | *Spathoglottis gracilis* Rolfe ex Hook.f. | FAN094 | Kg. Liposu, Ranau, Sabah, Malaysia | MG868992 | MG869026 | MG869060 |
| 13. | *Spathoglottis hardingiana* C.S.P.Parish & Rchb.f. | FAN016 | Gunung Baling, Kedah, Malaysia | MG868993 | MG869027 | MG869061 |
| 14. | *Spathoglottis hardingiana* C.S.P.Parish & Rchb.f. | FAN056 | G. Pong, Kenering, Perak, Malaysia | MG868994 | MG869028 | MG869062 |
| 15. | *Spathoglottis hardingiana* C.S.P.Parish & Rchb.f. | FAN105/ K20160013 | Pulau Timun, Langkawi, Kedah, Malaysia | MG868995 | MG869029 | MG869063 |
| 16. | *Spathoglottis kimballiana* Hook.f. | FAN085 | Ranau, Sabah, Malaysia | MG868996 | MG869030 | MG869064 |
| 17. | *Spathoglottis kimballiana* var. *angustifolia* Ames | FAN076 | Bidu-Bidu FR, Telupid, Sabah, Malaysia | MG868997 | MG869031 | MG869065 |

**Table 1.** *Cont.*

| | Species | Voucher Number | Locality | GenBank Accession Number | | |
|---|---|---|---|---|---|---|
| | | | | ITS | *mat*K | *trn*L-F |
| 18. | *Spathoglottis kimballiana* var. *angustifolia* Ames | FAN104 | Sungai Tongod, Telupid, Sabah, Malaysia | MG868998 | MG869032 | MG869066 |
| 19. | *Spathoglottis kimballiana* var. *kimballiana* | FAN067 | Gunung Kinabalu, Ranau, Sabah, Malaysia | MG868999 | MG869033 | MG869067 |
| 20. | *Spathoglottis kimballiana* var. *kimballiana* | FAN092 | Kota Belud, Sabah, Malaysia | MG869000 | MG869034 | MG869068 |
| 21. | *Spathoglottis kimballiana* var. *kimballiana* | FAN093 | Pekan Nabalu, Ranau, Sabah, Malaysia | MG869001 | MG869035 | MG869069 |
| 22. | *Spathoglottis microchilina* Kraenzl. | FAN083 | Kinabalu Park Research Centre, Sabah, Malaysia | MG869003 | MG869037 | MG869071 |
| 23. | *Spathoglottis microchilina* Kraenzl. | FAN086 | Mamut Copper Mine, Sabah, Malaysia | MG869004 | MG869038 | MG869072 |
| 24. | *Spathoglottis microchilina* Kraenzl. | FAN091 | Sg. Lohan, Ranau, Sabah, Malaysia | MG869005 | MG869039 | MG869073 |
| 25. | *Spathoglottis parviflora* Kraenzl. | FAN061 | Wamena, Irian Jaya | MG869006 | MG869040 | MG869074 |
| 26. | *Spathoglottis plicata* Blume | FAN001 | Gunung Ledang, Johor, Malaysia | MG869007 | MG869041 | MG869075 |
| 27. | *Spathoglottis plicata* Blume | FAN063 | Lata Chemerong, Dungun, Terengganu, Malaysia | MG869008 | MG869042 | MG869076 |
| 28. | *Spathoglottis plicata* Blume | FAN090 | Puncak Post, Mamut Copper Mine, Sabah, Malaysia | MG869009 | MG869043 | MG869077 |
| 29. | *Spathoglottis plicata* Blume | FAN103 | Long Baleh, Sarawak, Malaysia | MG869010 | MG869044 | MG869078 |
| 30. | *Spathoglottis plicata* var. *alba* | FAN045 | Lata Tembakah, Terengganu, Malaysia | MG869011 | MG869045 | MG869079 |
| 31. | *Spathoglottis pubescens* Lindl. | FAN068 | Fang District, Chiang Mai, Thailand | MG869012 | MG869046 | MG869080 |
| 32. | *Spathoglottis unguiculata* (Labill.) Rcbh.f. | FAN024 | Isle of Pines, New Caledonia | MG869013 | MG869047 | MG869081 |
| 33. | *Tainia paucifolia* (Breda) J.J.Sm. | FAN597 | Taman Rimba Kenong, Jerantut, Pahang, Malaysia | MG869014 | MG869048 | MG869082 |
| 34. | *Calanthe tankervilleae* (Banks) M.W.Chase, Christenh. & Schuit. | FAN707 | Kundasang, Sabah, Malaysia | MG869015 | MG869049 | MG869083 |

### 2.3. DNA Extraction

Total genomic DNA was extracted from fresh or silica gel dried leaf materials following the 2 × Cetyl Trimethylammonium Bromide (CTAB) method with some modifications [22]. The modifications were applied especially when to treat the over-dried leaves. The modifications are: (1) the amount of polyvinyl-pyrrolidone (PVP-40T) was increased to 2% (*w/v*); equal to the amount of CTAB used, (2) the chloroform: isoamyl alcohol (24:1) were added three times instead of two, and (3) after adding the ice-cold propan-2-ol (isopropanol), the samples were kept in the freezer at −20 °C for two days. For *Spathoglottis*, DNA with good quality and quantity was obtained based on this modified method.

### 2.4. Amplification

The entire nrITS region was amplified using 17SE and 26SE primer set [23]. This region is approximately 800 base pair (bp) and includes ITS1, ITS2 and the 5.8S ribosomal gene. Partial *mat*K, approximately 930 bp in length, was amplified using primer pairs 360F and 1326R [24]. This set of primers were chosen as they give more than 80% sequencing success; and show better sequence quality and high discriminatory. Meanwhile for *trn*L-F, the region was amplified using the universal forward and reverse primers set (*c* and *f*) [25]. This region is approximately 1000 bp long and includes the *trn*L intron and *trn*L-*trn*F spacer (Table 2).

**Table 2.** Primers information for polymerase chain reaction amplification of nrITS, *mat*K, and *trn*L-F regions.

| Region | Primers Sequences (5′ to 3′) | Primer Name | Primer Length | PCR Product Size |
|---|---|---|---|---|
| ITS | F: ACGAATTCATGGTCCGGTGAAGTGTTCG<br>R: TAGAATTCCCCGGTTCGCTCGCCGTTAC | 17SE<br>26SE | 28 bp<br>28 bp | 800–1000 bp |
| *mat*K (partial sequence) | F: CGATCTATTCATTCAATATTTC<br>R: TCTAGCACACGAAAGTCGAAGT | 360F<br>1326R | 22 bp<br>22 bp | 900 bp |
| *trn*L-F | F: CGAAATCGGTAGACGCTACG<br>R: ATTTGAACTGGTGACACGAG | *c*<br>*f* | 20 bp<br>20 bp | 1000 bp |

The polymerase chain reaction (PCR) protocol used to amplify ITS, *mat*K and *trn*L-F was as follows: initial denaturation at 94 °C for 5 min (2 min for ITS), denaturation at 94 °C for 1 min (30 cycles for ITS, 35 cycles for *mat*K, 32 cycles for *trn*L-F), 1–1.5 min annealing (58 °C for ITS, 48 °C for *mat*K, 56 °C for *trn*L-F), 1 min elongation at 72 °C (3 min for ITS), and followed by a final elongation period of 7 min at 72 °C. PCR was performed on MyCycler Thermal Cycler (Bio-Rad, Hercules, CA, USA). The products were purified using Wizard® SV Gel and PCR Clean-Up System Kit (Promega, Madison, WI, USA) according to the manufacturer's instructions.

For *Spathoglottis*, amplifications of *mat*K were rather straight forward and easy. However, the process was somewhat complex and deliberated for the ITS and *trn*L-F regions. Bovine serum albumin (BSA) was added into reaction volumes of difficult templates (GC-rich or containing secondary structures) or samples with low PCR yields. BSA helps to increase PCR products from low purity templates and improves its availability for hybridization. For ITS, BSA is required in preventing the misamplification of ITS regions of other organisms (exp. Fungi, thus producing the multiple bands).

### 2.5. Sequence Editing and Alignment

All the sequences obtained from both sense and antisense strands of ITS, *mat*K, and *trn*L-F sequences were assembled to produce contig sequences using BioEdit ver. 7.2.5 [26]. Sequences with high noise were checked through for a second time by eye using the chromatogram of the sequences. Multiple alignments of all sequences from each gene region were performed using CLUSTAL W in MEGA ver. 6.06 by default settings [27]. Again, the alignments were checked by eye and manually adjusted. All *Spathoglottis*

nucleotide sequences obtained in this study were submitted and deposited in GenBank Nucleotide database (NCBI).

### 2.6. Database Search—BLAST

To ensure sequence similarities and all are *Spathoglottis* species, the sequences obtained were assessed using BLAST (Basic Local Alignment Search) through the National Center for Biotechnology Information (NCBI) databases (https://blast.ncbi.nlm.nih.gov/Blast.cgi (accessed on 1 July 2018)). When compared to the available *Spathoglottis* sequences in NCBI, most of the sequences from this study showed query coverage of more than 90% with E-value = zero. Therefore, sequences with less than 90% query coverage were not used for phylogenetic data analysis.

### 2.7. Maximum Likelihood Analysis

The maximum likelihood (ML) analysis was performed in MEGA ver. 6.06 [27]. The best substitution model was determined using the same software, implemented as the Find Best DNA/Protein Models (ML) Test [28]. The ML method was performed to find the optimal tree using a heuristic search strategy with 1000 replicates of random taxon addition, subtree-pruning-regrafting (SPR) branch swapping, and gaps treated as missing data. Starting tree was generated automatically by applying NJ/BioNJ algorithms. Levels of group support were estimated with 1000 bootstrap replicates. Groups were retained when bootstrap percentages (BS) ≥50%. Bootstrap percentages (BS) of 50–70% considered as weak; 71–85% as moderate and >85% as strong. Likelihood analyses for each of the gene region and combinations of gene regions were run separately.

### 2.8. Maximum Parsimony Analysis

Parsimony analysis for each of the gene region and combinations of gene regions were run separately in MEGA ver. 6.06 [27]. To find the most parsimonious trees, Maximum Parsimony (MP) analysis was performed using a heuristic search with 1000 replicates of random taxon addition, tree rearrangements using Tree-Bisection-Reconnection (TBR) branch swapping, holding ten trees per replicate (to minimize time spent on searching for large numbers of trees), and gaps treated as missing data. All characters were treated as unordered and equally weighted [29]. Internal support for clades/groups was assessed by bootstrapping with 1000 replicates and groups were retained when bootstrap percentages (BS) ≥50%.

### 2.9. Bayesian Inference Analysis

The Bayesian inference (BI) of phylogeny was generated using MrBayes 3.1.1 in TOPALi ver. 2.5 software [30,31]. Each of the gene regions and combinations of gene regions were treated as a separate partition. Model selection was made according to the Bayesian inference criterion (BIC), Akaike information criterion (AIC), and the likelihood ratio tests (LRTs) with 1000 bootstrapping threshold run [32]. The model with the lowest BIC score was chosen as the optimal model of sequence evolution. Each analysis consisted of two independent runs with four chains for 1,000,000 generations, sampling one tree in every 100th generation. The first 25% of the trees was discarded as the burn-in period, and the remaining trees were used to assess the posterior probabilities (PP) in a majority-consensus tree. Because posterior probabilities in Bayesian analysis are not equivalent to bootstrap percentages (as PP are generally much higher), criteria equivalent to the standard statistical test was used [33]. Groups with PP > 95% were considered as strongly supported, PP 90–95% as moderately supported, and PP < 90% as weakly supported.

### 2.10. Test of Incongruence

To evaluate congruence between different DNA regions (ITS, *mat*K, *trn*L-F), each dataset was analyzed independently using the Likelihood Ratio Tests (LRTs) to see if they produce similar topology [32]. The ILD (Incongruence Length Difference) test was also

conducted using PAUP\* 4.0b10 to assess heterogeneity of the different gene regions (nuclear and plastid), and to determine whether the gene regions can be combined [34,35]. If the ILD *p*-value is ≥0.05, it indicates that the two regions are congruent thus can be combined. The combination of nuclear and plastid gene regions enhanced the phylogenetic signal in the data and increased cladogram resolution.

## 3. Results

A total of 34 individual sequences representing 11 species and three infraspecific taxa of *Spathoglottis*, including two outgroup species were analyzed in this present work; of which two species were of Thailand (Indochina) origin (*S. eburnea* and *S. pubescens*), four species and an infraspecies from Peninsular Malaysia (*S. affinis*, *S. hardingiana*, *S. aurea*, *S. plicata* and *S. plicata* var. *alba*), three species and two infraspecific taxa from Borneo (*S. gracilis*, *S. kimballiana*, *S. kimballiana* var. *angustifolia*, *S. kimballiana* var. *kimballiana* and *S. microchilina*), *S. parviflora* from Irian Jaya (East Malesia) and the New Caledonian species, *S. unguiculata*. *Tainia paucifolia* and *Calanthe tankervilleae* were used as outgroup species to root the phylogenies of each gene region.

### 3.1. Groupings of Taxa Based on Selected Morphological Characters

Table 3 shows the groupings of 11 species and three infraspecific taxa of *Spathoglottis* based on their plant size, flower size and shape of labellum.

**Table 3.** The 11 species and three infraspecific taxa of *Spathoglottis* grouped based on the plant sizes (Dwarf *Spathoglottis* = ≤30 cm tall and Large *Spathoglottis* = ≥30 cm up to 2 m tall), flower sizes (small–sized = 3.0 cm cross, medium–sized = 3.0–4.0 cm cross and large–sized = >5.0 cm cross) and shape of labellum (narrow or broad/bilobulate).

| | Taxa | Plant Size | Flower Size | Shape of Labellum |
|---|---|---|---|---|
| 1. | *S. affinis* | Dwarf | Small | Broad/Bilobulate |
| 2. | *S. eburnea* | Dwarf | Small | Broad/Bilobulate |
| 3. | *S. hardingiana* | Dwarf | Small | Broad/Bilobulate |
| 4. | *S. pubescens* | Dwarf | Small | Broad/Bilobulate |
| 5. | *S. aurea* | Large | Medium | Narrow |
| 6. | *S. gracilis* | Large | Large | Broad/Bilobulate |
| 7. | *S. kimballiana* | Large | Medium | Broad/Bilobulate |
| 8. | *S. kimballiana* var. *angustifolia* | Large | Large | Broad/Bilobulate |
| 9. | *S. kimballiana* var. *kimballiana* | Large | Medium | Broad/Bilobulate |
| 10. | *S. microchilina* | Large | Medium | Narrow |
| 11. | *S. parviflora* | Large | Medium | Broad/Bilobulate |
| 12. | *S. plicata* | Large | Medium | Broad/Bilobulate |
| 13. | *S. plicata* var. *alba* | Large | Medium | Broad/Bilobulate |
| 14. | *S. unguiculata* | Large | Medium | Broad/Bilobulate |

### 3.2. Phylogenetic Analysis Based on ITS Data Matrix

The ITS data matrix of 34 taxa, two of which were the outgroups, comprised 872 nucleotide characters (including gaps); of which 594 characters (68.1%) were conserved among all taxa, 186 characters (21.3%) were parsimony informative, and 92 characters (10.6%) were non-informative. The ITS1 and ITS2 regions showed variable sequence lengths and G+C content (%). Boundaries of ITS1, ITS2 and 5.8S were determined from the published ITS sequences of *S. pubescens* in NCBI (Accession No: KM25162.1, KP51403.1, and KP51405.1); and start codons for translation initiation for 5.8S gene were assessed by referring to a validly published report [36].

The sequence lengths of ITS1 for all 11 species and three infraspecific taxa of *Spathoglottis* ranged from 244–260 bp, while ITS2 sequence lengths ranged from 318–345 bp (Table 4). In contrast, the length of 5.8S was uniform for all *Spathoglottis* species (157 bp). The G+C content (%) of ITS2 was found to be slightly higher compared to ITS1; with average G+C content of 64.4% and 57.6% were recorded for ITS2 and ITS1, respectively. The 5.8S region has been found to be more conserved as evidenced from the number of conserved sites (140/157 = 89.2%), followed by ITS2 (60.3%), and ITS1 (53.7%).

**Table 4.** Sequence data analysis of ITS1, 5.8S and ITS2 regions in 11 species and three infraspecific taxa of *Spathoglottis*.

| Region | Aligned Nucleotide Length (bp) | Conserved Site (%) | Variable Site(%) | Parsimony Informative Site (%) | G+C Content (%) |
|---|---|---|---|---|---|
| ITS1 | 244–260 | 137 (53.7) | 116 (45.5) | 85 (33.3) | 57.6 |
| 5.8S | 157 | 140 (89.2) | 17 (11) | 10 (6.4) | 56.8 |
| ITS2 | 318-345 | 208 (60.3) | 125 (36.2) | 88 (25.5) | 64.4 |

The average percentage of sequence divergence within *Spathoglottis* was 6.8% with highest percent divergence between species was shown to be between *S. affinis* and *S. unguiculata* (13.8%). The average base compositions among all taxa analyzed were A = 22.8%, C = 26.6%, T = 18.7%, and G = 20.21%. The estimated transition/transversion bias (*R*) was 3.28.

Figure 2 shows a combined tree of ML and Bayesian analysis as both trees shared similar tree topology. For the ML analysis, there was a total of 780 positions in the final dataset with highest log likelihood = (−2833.908).

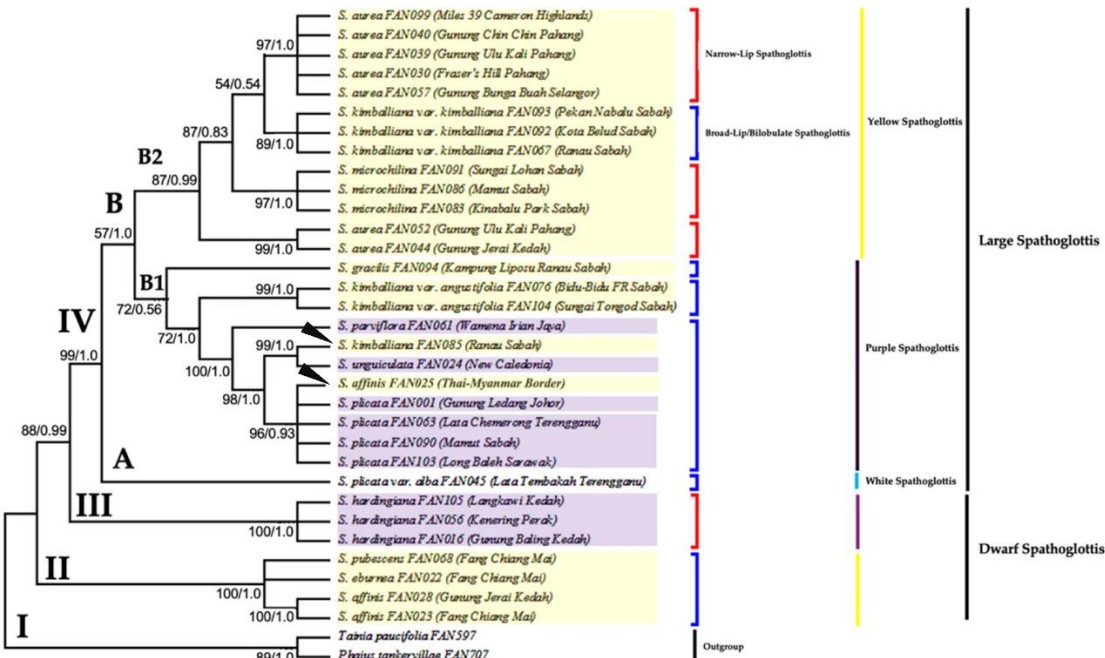

**Figure 2.** The combined majority-rule consensus tree results from ML analysis and optimal tree results from Bayesian analysis generated from the ITS sequence dataset. Number below branches are bootstrap percentages ≥ 50 and probability values ≥ 0.5 (BS/PP). Clade II–IV shows the ingroups of genus *Spathoglottis*; Clade I indicates outgroup. Clade II and III holds the Dwarf *Spathoglottis* species, while Clade IV (A and B) consists of the Large *Spathoglottis* species. Clade B is further separated into

subclades B1 and B2; each holds the Large Purple and Large Yellow *Spathoglottis* species group, respectively. Taxa highlighted in yellow box are the yellow flower *Spathoglottis* while purple box comprises of flowers in different shades of purple. Groupings of taxa were also based on the shape of the labellum; either narrow or broad/bilobulate.

The unweighted parsimony analysis resulted in seven most parsimonious trees. The strict-consensus tree is shown in Figure 3. Bootstrap percentages (BS) of parsimonious trees in which the associated taxa clustered together in the bootstrap test (1000 replicates) are shown next to the branches.

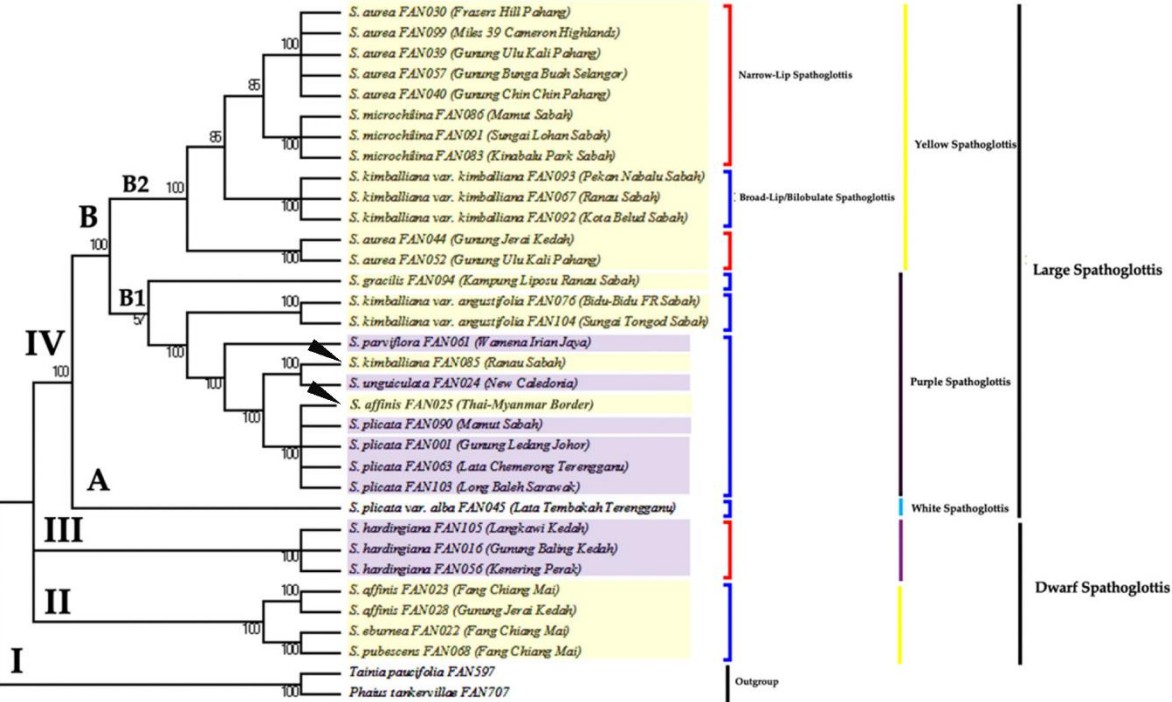

**Figure 3.** The consensus tree inferred from seven equally parsimonious trees via MP analysis of genus *Spathoglottis* based on ITS sequence. Number below branches are bootstrap percentages ≥ 50. Branches corresponding to partitions reproduced in less than 50% trees are collapsed. Tree length = 346; consistency index, CI = 0.712; retention index, RI = 0.909; and composite index, RC = 0.699 (RC = 0.647) for all sites and parsimony-informative sites (in parentheses). Clade II–IV shows the ingroups of genus *Spathoglottis*; Clade I indicates outgroup. Clade II and III holds the Dwarf *Spathoglottis* species, while Clade IV (A and B) consists of the Large *Spathoglottis* species. Clade B is further separated into subclades B1 and B2; each holds the Large Purple and Large Yellow *Spathoglottis* species group, respectively.

The topology of the ML, MP and Bayesian trees inferred from the ITS region were significantly congruent together. Four major clades (I–IV) were identified (Figures 2 and 3) based on the following criteria: (1) that they are well-supported and monophyletic, and (2) that they are morphologically distinguishable. Based on the ITS trees, species under a clade were grouped together according to the: (1) plant size (dwarf or large-sized); (2) flower size (small, medium, or large); (3) flower colour (purple or yellow); and 4) lip/midlobe shape (narrow or broad/bilobulate). The monophyly of the clades was supported by strong (BS) and (PP) values.

### 3.3. Phylogenetic Analysis Based on Combined Plastid Sequence Data

The ILD test between combined plastid regions gave a *p*-value of 0.10; showing significant congruence among the partitions of the dataset, hence was subjected to a combined analysis. The aligned sequences of the combined plastid dataset consisted of 2373 nucleotide characters (with gaps), of which 1870 (78.8%) were conserved characters, 337 characters

(14.2%) were parsimony informative, and 166 characters were non-informative (7.0%). The average sequence divergence for *Spathoglottis* species from the combined plastid data matrix was 2.8%. Species with the highest percent of divergence was shown between *S. affinis* and *S. unguiculata* (5.4%).

Figure 4 shows a combined tree of ML and Bayesian analysis as both trees shared similar tree topology. The ML tree has 1000 replicates with highest log likelihood (−4503.460).

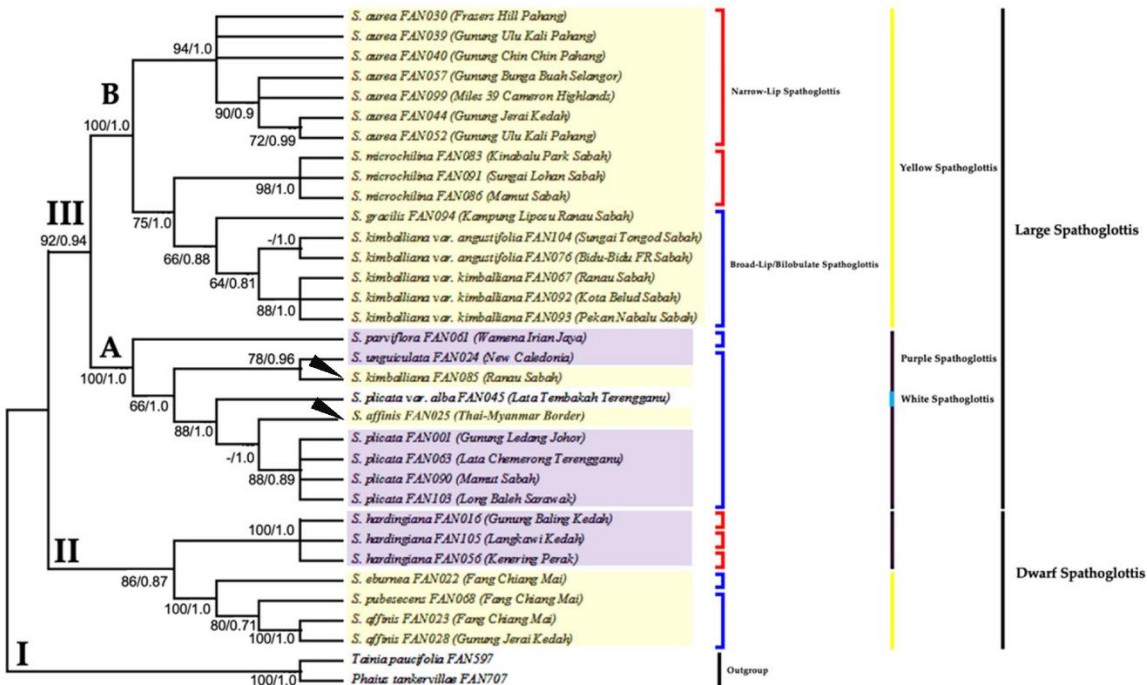

**Figure 4.** The combined majority-rule consensus tree results from ML analysis and optimal tree results from Bayesian analysis generated from the combined plastid gene dataset. Number below branches are bootstrap percentages ≥ 50 and probability values ≥ 0.5 (BS/PP). Clade II and III holds all 11 species and three infraspecific taxa (ingroups) of *Spathoglottis*; Clade I indicates outgroup. Clade II comprises of the Dwarf *Spathoglottis* species, while Clade III consists of the Large *Spathoglottis* species. Clade III is further separated into subclades A and B; the Large Purple and Large Yellow *Spathoglottis* groups, respectively. Groupings of taxa were also based on the shape of the labellum; either narrow or broad/bilobulate. Both clades of Dwarf and Large *Spathoglottis* (Clade II and Clade III) are monophyletic.

Meanwhile, the unweighted parsimony analysis resulted in four most parsimonious trees with tree length = 337, consistency index (CI) = 0.780, retention index (RI) = 0.935, and composite index, RC = 0.799 (0.729) for all sites and parsimony-informative sites (in parentheses). The strict-consensus tree is shown in Figure 5.

This combined plastid data analysis has produced a better result with improved tree resolutions, than analyses with each plastid marker individually. The trees were topologically similar to the *mat*K trees but with better branch support, and show no contradiction to the *trn*L-F trees. All the ML, MP and Bayesian trees of the combined plastid markers were congruent to each other with high (BS) and (PP) support values.

There were three major clades in the combined plastid trees (Figures 4 and 5). Clade II-III holds all the 11 species and three infraspecific taxa of *Spathoglottis*, while Clade I indicates outgroup. Clade II consists of the two Dwarf *Spathoglottis* groups, the Dwarf Yellow *Spathoglottis* and the Dwarf Purple *Spathoglottis.* This clade is strongly supported in both ML and MP trees with $BS_{ML}$ = 86%, $BS_{MP}$ = 100%; but show moderate branch support in the Bayesian tree ($PP_B$ = 0.87). The Dwarf *Spathoglottis* species formed the basal groupings. Species within Clade II were also finely separated according to the colour of the flowers and shape of the lip. The Dwarf Yellow *Spathoglottis* clade ($BS_{ML}$ = 100%, $BS_{MP}$ = 100%,

PP$_B$ = 1.0) that housed together all the broad-lip/bilobulate species formed a distinct clade from its narrow-lip sister, *S. hardingiana* in the Dwarf Purple clade (BS$_{ML}$ = 100%, BS$_{MP}$ = 100%, PP$_B$ = 1.0). Clade II is monophyletic.

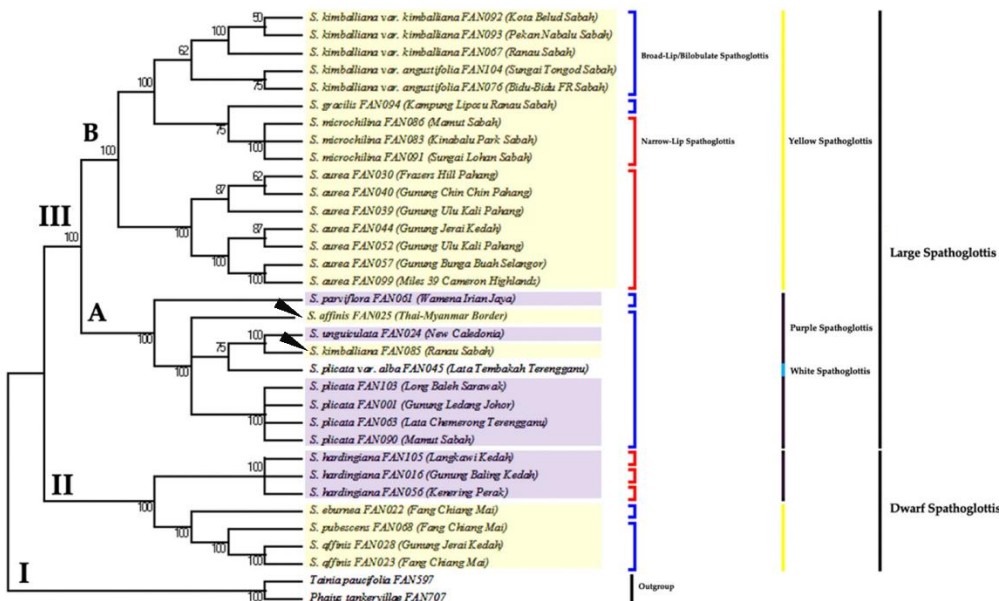

**Figure 5.** The consensus tree inferred from eight equally parsimonious trees via MP analysis of genus *Spathoglottis* based on combined plastid gene dataset. Number below branches are bootstrap percentages ≥ 50. Branches corresponding to partitions reproduced in less than 50% trees are collapsed. Clade II and III holds all the ingroups; Clade I indicates outgroup. Clade II comprises of the Dwarf *Spathoglottis* species, while Clade III (A and B) consists of the Large *Spathoglottis* species.

All the large species of *Spathoglottis* were nested within Clade III (Large *Spathoglottis* group). This clade is monophyletic and is well-supported in all trees with (BS$_{ML}$= 92%, BS$_{MP}$ = 100%, PP$_B$ = 0.94). Clade III is further divided into Subclade III-A and Subclade III-B, the Large Purple *Spathoglottis* group and the Large Yellow *Spathoglottis* group, respectively. Both subclades obtained maximum branch supports with BS$_{ML}$ = 100%, BS$_{MP}$ = 100%, and PP$_B$ = 1.0. Positions of the purple-flower *Spathoglottis* within Subclade III-A were also much resolved. However, from this combined analysis, the Large Purple *Spathoglottis* group (Subclade III-A) revealed to be polyphyletic due to the unsolved placements of the two yellow-flower species (*S. affinis*_FAN 025 and *S. kimballiana*_FAN085, marked in figures) embedded within the group.

Meanwhile, subclade III-B (Large Yellow *Spathoglottis* group) is monophyletic, and the clade is fully supported by strong (BS) and (PP) values (BS$_{ML}$ = 100%, BS$_{MP}$ = 100%, PP$_B$ = 1.0). The yellow-flower species within this clade is further divided into two subgroups based on their geographical distribution and geological history. The *S. aurea* clade is strongly-supported with BS$_{ML}$ = 94%, BS$_{MP}$ = 100%, and PP$_B$ = 1.0. *Spathoglottis aurea* is then further grouped according to their pollination strategies; between the cleistogamous flower group (BS$_{ML}$ = 93%, BS$_{MP}$ = 87%, PP$_B$ = 1.0) and the geitonogamous flower group (BS$_{ML}$ = 90%, BS$_{MP}$ = 100%, PP$_B$ = 0.9). For the Bornean *Spathoglottis* clade (BS$_{ML}$ = 75%, BS$_{MP}$ = 100%, PP$_B$ = 1.0), it is very obvious that *S. gracilis* and *S. microchilina* that possessed broad-plicate leaf were separated from the two narrow-grassy leaf infraspecies, *S. kimballiana* var. *angustifolia* and *S. kimballiana* var. *kimballiana*. In the MP tree, *S. gracilis* was observed to be genetically close to the narrow-lip *S. microchilina* (BS$_{MP}$ = 75%), but is sister to *S. kimballiana* var. *kimballiana* and *S. kimballiana* var. *angustifolia* in the ML and Bayesian tree (BS$_{MP}$ = 66%), PP$_B$ = 0.8).

This combined plastid data analysis has provided a better insight into the relationship among species of the main four groups of *Spathoglottis*. The cladograms were well-resolved with better resolution and phylogenetic signals. However, the positions of *S. affinis* (FAN025)

and *S. kimballiana* (FAN085) within the Large Purple *Spathoglottis* group require further clarification.

### 3.4. Phylogenetic Analysis Based on Combined Plastid and nrITS Data

Combination of data analysis of the nuclear and plastid gene regions will enhance the phylogenetic signal in the data and increased cladogram resolution. The combined data analyses provided better insight into species delimitation within the four main groupings of *Spathoglottis* diagnosed in this present work.

The probability value obtained from the ILD test between plastid DNA and nrITS dataset (*p*-value = 0.14) indicated that the two matrices could be combined for analysis. The combined alignment of all plastid markers and nrITS consisted of 3253 nucleotide characters (with gaps), of which 2489 characters (76.5%) were conserved, 497 characters (15.3%) were parsimony informative, and 267 (8.2%) characters were non-informative. The average sequence divergence for *Spathoglottis* species from the combined plastid and nuclear data matrix was 3.9%. The average base compositions among all taxa analyzed were A = 29.8%, C = 19.2%, T = 31.3%, and G = 19.7%. The estimated Transition/Transversion bias (*R*) was 1.44.

Figure 6 shows a combined tree of ML and Bayesian analysis as both trees shared similar tree topology. The ML tree has 1000 replicates with highest log likelihood (−4503.460). There was a total of 1765 positions in the final dataset with highest log likelihood= (−7663.795).

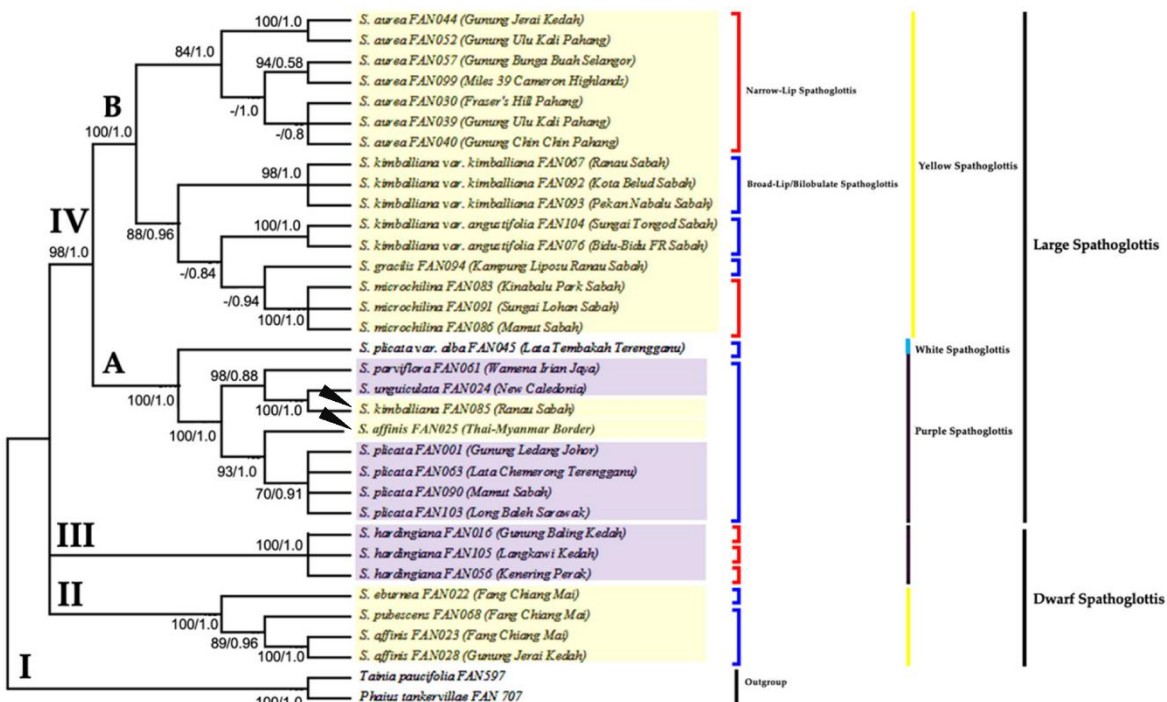

**Figure 6.** The combined majority-rule consensus tree results from ML analysis and optimal tree results from Bayesian analysis generated from the combined plastid and nuclear gene dataset. Number below branches are bootstrap percentages ≥ 50 and probability values ≥ 0.5 (BS/PP). Clade II–IV holds all ingroups of genus *Spathoglottis*; Clade I indicates outgroup. The Dwarf Yellow *Spathoglottis* (Clade II) formed a basal group; while the Dwarf Purple *Spathoglottis* (Clade III) is sister to all Large *Spathoglottis* species. Clade IV that comprises of the Large *Spathoglottis* groups is monophyletic. Independently, the Large Yellow *Spathoglottis* group (Subclade IV-B) is monophyletic while the Large Purple *Spathoglottis* group (Subclade IV-A) is polyphyletic.

Meanwhile, the unweighted parsimony analysis resulted in four most parsimonious trees. The strict-consensus tree is shown in Figure 7.

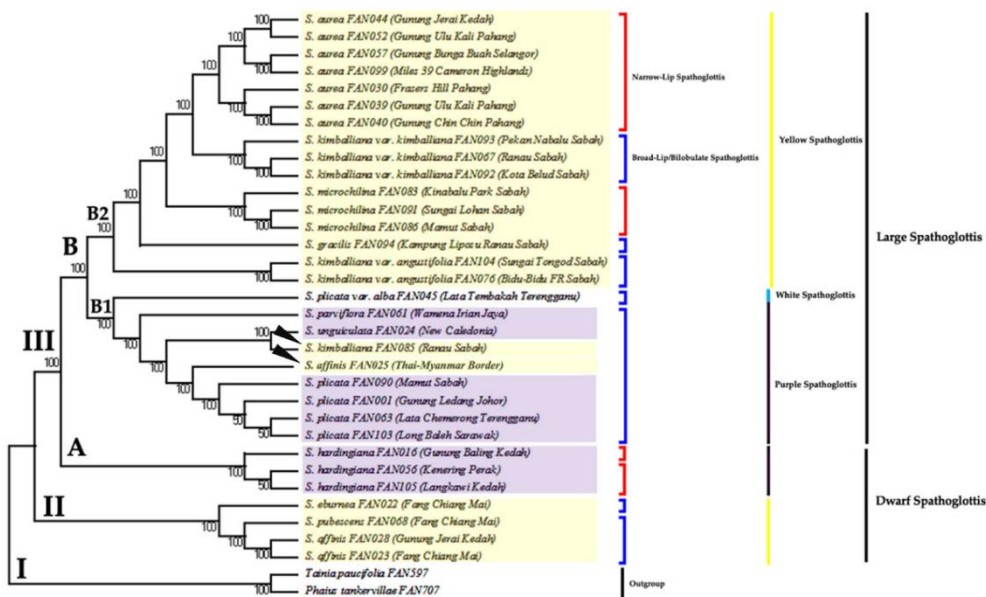

**Figure 7.** The consensus tree inferred from two equally parsimonious trees via MP analysis of genus *Spathoglottis* based on combined plastid and nuclear gene dataset. Number below branches are bootstrap percentages ≥ 50. Branches corresponding to partitions reproduced in less than 50% trees are collapsed. Tree length = 695; consistency index, CI = 0.709; retention index, RI = 0.906; and composite index, RC = 0.711 (0.643) for all sites and parsimony-informative sites (in parentheses). Clade I indicates outgroup. Clade II comprises of the Dwarf Yellow *Spathoglottis* group, as sister to all *Spathoglottis* species. Clade III holds the Dwarf Purple *Spathoglottis* group (Subclade III-A), Large Purple *Spathoglottis* group (Subclade III-B1) and Large Yellow *Spathoglottis* group (Subclade III-B2). The Large *Spathoglottis* group (Clade III-B) is monophyletic.

This combined analysis has produced well-resolved trees and species delimitation among the four groups of *Spathoglottis* was very clear. All the ML, MP, and Bayesian trees were congruent to each other. In the combined ML and Bayesian tree (Figure 6), there were four major clades. Clade I consists of the outgroup species while Clade II–IV hold all the ingroup *Spathoglottis* species. Clade II, the Dwarf Yellow *Spathoglottis* is sister to all *Spathoglottis* species with maximum branch supports $BS_{ML}$ = 100% and $PP_B$ = 1.0. The Dwarf Purple *Spathoglottis* formed Clade III, which was originally placed within Clade IV but was collapsed to a polytomy due to weak branch support ($BS_{ML}$ = 48%, $PP_B$ < 0.5).

Clade IV is monophyletic with strong (BS) and (PP) supports ($BS_{ML}$ = 98%, $PP_B$ = 1.0), and was further divided into two subclades according to the colour of the flowers. Subclade IV-A ($BS_{ML}$ = 100%, $PP_B$ = 1.0) consists of all the Large Purple *Spathoglottis* species and is polyphyletic; due to the placements of the two yellow-flower *Spathoglottis* (*S. kimballiana*_FAN085 and *S. affinis*_FAN025) within this clade. Eventhough *S. kimballiana* has distinct morphological attributes and is geographically isolated from *S. unguiculata*, this two species were connected by having more than four leaves per plant and possessed tough coriaceous sepals; characteristics unique to this two species.

Meanwhile, subclade IV-B that comprises of all the large-yellow *Spathoglottis* species is monophyletic ($BS_{ML}$ = 100% and $PP_B$ = 1.0). The *S. aurea* clade ($BS_{ML}$ = 84% and $PP_B$ = 1.0) from Peninsular Malaysia is well separated from the Bornean yellow-flower *Spathoglottis* ($BS_{ML}$ = 88% and $PP_B$ = 0.96). *Spathoglottis aurea* were then further grouped based on their flower pollination strategies. The open-flower or insect pollinated *S. aurea* from Gunung Jerai and Gunung Ulu Kali were well separated from their self-pollinated sisters. While within the Bornean *Spathoglottis* clade, *S. gracilis* was observed to be genetically close to *S. microchilina* and *S. kimballiana* var. *angustifolia*. *Spathoglottis kimballiana* var. *angustifolia*, a lowland species with large flower is sister to all montane *Spathoglottis* species possessed medium-sized flower. *Spathoglottis microchilina*, *S. kimballiana* var. *kimballiana* and *S. aurea*

shared the characteristic of having short sidelobes as to compare to the large and auriculate one in *S. kimballiana* var. *angustifolia* and *S. gracilis*.

While for the MP analysis, the tree produced was topologically similar to those of ML and Bayesian's but was much resolved (Figure 7). The Dwarf Yellow *Spathoglottis* formed Clade II ($BS_{MP}$ = 100%) as a basal group. Clade III is strongly supported with $BS_{MP}$ = 100%, and it holds together the Dwarf Purple *Spathoglottis* (Clade III-A; $BS_{MP}$ = 100%) and the Large *Spathoglottis* group (Clade III-B; $BS_{MP}$ = 100%). From this MP analysis, it was suggested that the Dwarf *S. hardingiana* is sister to all large *Spathoglottis* species; whereas its position in the ML and Bayesian trees is rather equivocal. Clade III-A was then divided into two subclades which are Subclade III-B1 (Large Purple *Spathoglottis* group) and Subclade III-B2 (Large Yellow *Spathoglottis* group). Both subclades are strongly supported with $BS_{MP}$ = 100%. The arrangements of species within the Large Purple *Spathoglottis* group were also similar to those of in combined ML and Bayesian tree. While for the Large Yellow *Spathoglottis* group, the positions of each of the species in this clade were much resolved. The lowland and ecological peculiar *Spathoglottis kimballiana* var. *angustifolia* is sister to the rest of yellow-flower montane species. The montane species were consisted of *S. gracilis*, *S. microchilina*, *S. kimballiana* var. *kimballiana* and *S. aurea*; and the clade is supported with strong BS value ($BS_{MP}$ = 100%). *Spathoglottis gracilis*, another large-flower species is sister to the medium-sized species within the montane *Spathoglottis* group.

From this combined plastid and nuclear analysis, the trees produced were much-resolved with better resolution, strong phylogenetic signals, and maximum clade credibility. Species discrimination was also best described with reduced polytomies. Meanwhile, the two much confused narrow-lip species; *S. aurea* and *S. microchilina* were proven to be distantly related and formed different clades separately.

## 4. Discussion

### 4.1. The Efficacy of Nuclear and Plastid Gene Regions in Inferring Phylogenetic Relationships

The independent gene analyses together with the combined plastid and nuclear dataset have successfully inferred the phylogenetic relationships of *Spathoglottis* in Peninsular Malaysia and Borneo. The positions and species boundaries among several debatable taxa were also well clarified and circumscribed. The results obtained from the ML analysis, MP analysis and Bayesian analysis of each of this gene region; coupled with the combined plastid and nuclear dataset have showed that *Spathoglottis* were constituted of four major groups. These groups are the Dwarf Yellow *Spathoglottis*, Dwarf Purple *Spathoglottis*, Large Yellow *Spathoglottis*, and Large Purple *Spathoglottis*. Previously, no attempt has been carried out to infer the relationships among species of *Spathoglottis*, be it based on morphological or molecular aims; thus, any formal sectionals within this genus have never been proposed. Hence, the groupings obtained from this study will be used as the basis for further classifications and subdivisions of this genus.

The nrITS is one of the loci that is commonly used for phylogenetic inference as it provides high copy numbers and relative range of phylogenetic utility [37]. The nrDNA gene is very heterogeneous both in size and nucleotide sequences among wide range of angiosperms [38]. The utility of this gene region in the molecular study of family Orchidaceae is very widespread and is used in various taxonomic levels; especially the low-level taxa such as infrageneric species [39–44]. From this present work, it was proven that the ITS region is a powerful tool in providing a clearer understanding of the phylogeny of a major part of genus *Spathoglottis.* A high level of parsimony informative characters (PIC = 21.3%) was established in this region, thus showing a high level of branch support in the data analysis. The clades were largely resolved, and species delimitation was well-clarified.

Due to its highly conserved region (91.6%), *mat*K can be easily amplified. Partial *mat*K sequences that were used in this present work were able to produce phylogenetic trees that were comparable in resolution and support of the trees. The efficient utility of *mat*K sequences as barcodes for identifying species and in inferring phylogeny has been demonstrated in many molecular studies among the family Orchidaceae [24,45–49]. Results

from the *mat*K trees from both independent and combined plastid data analysis showed that separations among the four *Spathoglottis* groups were good, with strong bootstrap and posterior probabilities supports. However, separation among species, particularly in the Large Yellow *Spathoglottis* and Large Purple *Spathoglottis* groups, were not as successful due to the internal branch polytomies (PIC = 4.8%).

The *trn*L-F region is another plastid marker that is widely used in the systematics of family Orchidaceae [41,46,50,51]. It has been recognized to have phylogenetic utility in a broad range of taxonomic level; from the family down to species, and occasionally to the levels of populations [37]. In this present work, the *trn*L-F region contains higher PIC values (19.5%) to compare to *mat*K. However, independent analysis of the *trn*L-F region has shown that the region was unable to resolve the relationships among the four groups of *Spathoglottis*, eventhough it may confirm the positions of species belonging to each of the group/clade. The *trn*L-F region is also variable enough to discriminate between species of *Spathoglottis*; exactly to the population level as observed in the *S. aurea* clades. Thus, the two plastid gene regions were combined to infer the best evolutionary relationships among species of *Spathoglottis* from Peninsular Malaysia and Borneo.

### 4.2. Monophyly of Genus Spathoglottis

Based on the analyses of both independent and combined dataset, the most important finding regarding the phylogenetic relationships of the four groups was: the Dwarf Yellow *Spathoglottis*, Dwarf Purple *Spathoglottis*, Large Yellow *Spathoglottis* and Large Purple *Spathoglottis* have demonstrated that although they were not all independently monophyletic, they formed well-supported monophyletic groups with strong supports. Each analysis has shown that, independently, the Dwarf Yellow *Spathoglottis* and Large Yellow *Spathoglottis* are monophyletic, while the Large Purple *Spathoglottis* is polyphyletic. In the combined plastid analysis, both the Dwarf Yellow *Spathoglottis* and Dwarf Purple *Spathoglottis* were joined together to form the monophyletic Dwarf *Spathoglottis* group. Meanwhile in the nrITS and combined plastid and nuclear analyses, it was revealed that the Dwarf Yellow *Spathoglottis* was sister to all *Spathoglottis* species while *S. hardingiana* (Dwarf Purple *Spathoglottis*) is sister to all large-flower *Spathoglottis.* In all analyses, the Large *Spathoglottis* group formed a strongly supported monophyletic group by holding both the Large Yellow *Spathoglottis* and Large Purple *Spathoglottis* species groups.

The groupings obtained from thes molecular analyses were well-supported and in accordance to the early revisions on the morphological classification of the genus [8,9]. Traditionally, species in *Spathoglottis* was classified based on their: (1) plant size, (2) flower size, (3) flower colour, (4) lip/midlobe shape, (5) sidelobe shape, and (6) flower-pollination strategies.

The Dwarf *Spathoglottis* group consists of the miniature *Spathoglottis* species (≤30 cm tall), which are mainly native to the Indochina region. However, two species escaped the boundary, occuring lower down towards the northernmost part of Peninsular Malaysia (*S. hardingiana* and *S. affinis*). From the finding of this present work, this Dwarf Group was then further separated into the Dwarf Purple *Spathoglottis* and Dwarf Yellow *Spathoglottis* groups based on the colour of the flowers, lip shapes, and flower resupination. *Spathoglottis hardingiana* from the Dwarf Purple *Spathoglottis* group is distinguishable from the rest of Dwarf Yellow *Spathoglottis* species by having a non-resupinated lip; in which the lip is on the uppermost part of the flower, rather than turning downwards to the normal position as observed in other *Spathoglottis* species. This trait is autapomorphic for *S. hardingiana.* In addition, *S. hardingiana* possessed a narrow lip and almost thread-like without distinct sidelobes; very contrasting to the broad lip and large sidelobes as in *S. affinis* (bilobulate), *S. eburnea* (square), and *S. pubescens* (obcordate).

Meanwhile, the Large *Spathoglottis* group is a monophyletic group that housed all the large-sized *Spathoglottis*; particularly referring to the size of the flowers. Almost all the species possessed broad-plicate leaves, except for the two infraspecies; *S. kimballiana* var. *kimballian*a and *S. kimballiana* var. *angustifolia* with the grass-like leaves. Within this Large

*Spathoglottis* group, the species were then further separated into two groups according to the colour of the flowers, number of leaves, floral bract shape, floral bract texture, sepals texture, lip size, sidelobe lengths, and geological attributes.

The flowers in different shades of purple and possessing a broad/bilobulate lip were placed together in the Large Purple *Spathoglottis* group. The white form of *S. plicata*; *S. plicata* var. *alba* was also embedded within this clade. Besides the colour of the flowers, the species of Large Purple *Spathoglottis* group was also characterized by having four or more than four leaves, bilobulate lip, elliptical and soft floral bract, hairy pedicel, thin sepals, short sidelobes, short lip, and generally the plant grows on multisubstrate habitat. However, eventhough the clade was strongly supported by the (BS) and (PP) values; it was showed from all separate gene regions and combined dataset analyses that the Large Purple *Spathoglottis* is polyphyletic. This is due to the positions of the two yellow-flower *Spathoglottis*, *S. kimballiana*_FAN085 and *S. affinis*_FAN025 within this clade. Ideally, *S. kimballiana* was supposedly to be placed in the Large Yellow *Spathoglottis* group while *S. affinis* nested within the Dwarf Yellow *Spathoglottis* Complex. These cases of morphological convergence have suggested that the flower colour trait is homoplastic.

The Yellow *Spathoglottis* group is a monophyletic group placed at the crown position of the phylogenetic trees, evidence of a recent radiation. It was well supported and nests together all the yellow-flower *Spathoglottis*, which were further separated according to various floral traits and ecological information such as flower size, lip shape, callus architecture, the content of anthocyanin, habitat niches, and geological attributes. Based on the separate plastid markers analyses coupled with the combined dataset, the Large Yellow *Spathoglottis* were split into the Peninsular Malaysia group (*S. aurea*) and the Bornean *Spathoglottis* group (*S. gracilis*, *S. microchilina*, *S. kimballiana* var. *kimballiana*, *S. kimballiana* var. *angustifolia*). However, this Bornean *Spathoglottis* group was weakly resolved in the independent nrITS, *mat*K, and *trn*L-F analysis; but largely solved in the combined analyses. Results obtained from this present work have also proven that the two most-conflicting narrow-lip *Spathoglottis* species; *S. aurea* and *S. microchilina* were discriminated as two different species. This was supported by the placements of each of the species in separate clades with strong support values.

*4.3. Proposed Taxonomic and Nomenclatural Changes for Spathoglottis plicata* var. *alba*

From the molecular analyses, it was observed that the white-flower *S. plicata* var. *alba* had showed a distant genetic divergence from the purple-flower *S. plicata.* Based on the ML, MP, and Bayesian trees of the nuclear, plastid and combined gene analyses; *S. plicata* var. *alba* formed a distinct clade of its own. To see how distant *S. plicata* var. *alba* is from *S. plicata*, the genetic difference between these two entities were calculated. It was shown that the genetic divergence among *S. plicata* var. *alba* and *S. plicata* was 7.8–9.0% for ITS, 0.2% for *mat*K, 0.3% for *trn*L-F, and 2.8–3.0% for the combined plastid and nuclear genes, respectively. To further support that either *S. plicata* var. *alba* can be elevated into a species rank, a separate DNA barcoding gap analysis was conducted. From the DNA barcoding gap analysis, it appears that the maximum boundaries, or referred to as the 'threshold or gap value' between different species in genus *Spathoglottis* were 1.4% for ITS2, 0.2% for *mat*K, 0.5% for *trn*L-F, and 0.5% for the combined plastid and nuclear genes. Thus, if the values of genetic differences showed between *S. plicata* var. *alba* and *S. plicata* were higher than those threshold values, taxonomic and nomenclatural changes should be considered for *S. plicata* var. *alba*.

Supported by both phylogenetic and DNA barcoding gap analyses, it was shown that the white *S. plicata* var. *alba* is genetically distant from *S. plicata.* Hence, it is proposed that the taxonomic status of *S. plicata* var. *alba*, be classified from a variety to a species rank.

**5. Conclusions**

The analyses of independent gene regions coupled with the combined dataset have proven that the genus *Spathoglottis* is monophyletic. The phylogenetic relationships among

the 11 species and three infraspecific taxa of *Spathoglottis* from Peninsular Malaysia and Borneo were successfully resolved, and species boundaries were successfully circumscribed. Four major groups were determined from these analyses: (1) Dwarf Purple *Spathoglottis*, (2) Dwarf Yellow *Spathoglottis*, (3) Large Purple *Spathoglottis*, and (4) Large Yellow *Spathoglottis*. From this study, it is proposed that the split in the Dwarf and Large *Spathoglottis* groups reflects an early differentiation of plant size, flower colors, and flower size. In addition, morphology alone can be misleading for inferring the relationships among groups of interest, as floral morphology is highly flexible.

**Author Contributions:** Data curation, F.A.N., A.S.O. and K.N.A.M.; formal analysis and investigation, F.A.N., A.S.O., K.S. and R.G.; methodology, F.A.N.; resources, A.S.O., K.S. and R.G.; supervision, A.S.O. and K.S.; validation, A.S.O.; writing—original draft, F.A.N.; writing—review and editing, F.A.N. and A.S.O. All authors have read and agreed to the published version of the manuscript.

**Funding:** The study was funded by the USM Short Term Grant of assignment No. 304/PBI-OLOGI/6315549 awarded to the corresponding author and the fifth author. All funders provided financial supports for the present work but did not contribute any additional role in in the research design, data collections and analysis, and preparation of the manuscript.

**Data Availability Statement:** Data presented in this article are available on request from the corresponding authors.

**Acknowledgments:** The authors would like to express our deepest gratitude to the administrative and field personnel of School of Biological Sciences, Universiti Sains Malaysia; Department of Biology, Universiti Putra Malaysia; and Institute for Tropical Biology and Conservation, Universiti Malaysia Sabah for the facilities and assistance provided during the study conducted. Thank you to Forestry Department of Peninsular Malaysia, Forestry Department of Sarawak, Forestry Department of Sabah, Sabah Biodiversity Centre and Kinabalu Park for the research permits granted. Many thanks to Richard Chung (FRIM), Ong Poh Teck (FRIM), Yong Kien Thai (KLU), Nik Norhazrina Nik Mohd Kamil (UKMB) for the assistance provided during visits to the herbaria, to our collaborators, and special tribute to the late Peter O'Byrne for his kind sharing of thoughts and beautiful photos.

**Conflicts of Interest:** The authors declare no conflict of interest.

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
