# Peer review of "Molecular Phylogenetics of the Orchid Genus Spathoglottis (Orchidaceae: Collabieae) in Peninsular Malaysia and Borneo"

_forests, doi:10.3390/f13122079_

Round 1

Reviewer 1 Report

Nordin et al used three markers to reconstruct the phylogeny of Spathoglottis. It would be better to use more markers or even plastome to do this to get robust results.

The plates are in poor quality and can be improved. The plates (Figure 2-9) are needed to improve. (1) what do the numbers below branch stand for? (2)The resolution of figre 1 to 9 is not very high. 

Author Response

Dear Reviewer,

Thank you so much for your constructive comments for the betterment of my manuscript. I have made changes to the original manuscript as suggested:

1) Figure 2-7 have been improved with better resolution.

2) Numbers below branch stand for the boostrap percentages (BS) probability   values (PP). The information has been included in each figure (Figure 3-7).

Thank you so much.

FAN, KS, RG, KNAM and ASO

Reviewer 2 Report

In this study, the nrITS, matK and trnL-F fragments were used to construct nuclear gene and chloroplast evolutionary trees, respectively, to study the phylogenetic relationships of the Orchid Genus Spathoglottis from Peninsular Malaysia and Borneo. It meets the urgent need that although Spathoglottis have great horticultural value, their comprehensive research is still blank. The monophyletic status of Spathoglottis was clarified, the phylogenetic relationships of 11 species and 3 subspecies taxa of Spathoglottis were elucidated, and the controversial taxonomic issues between the yellow-flower S.aurea and S.microchilina were successfully resolved. This study provided a theoretical support for the better use of Spathoglottis.

However, there are some problems, which must be resolved before it is considered for publication. If the following problems are well-addressed, this reviewer believes that the essential contribution of this paper is important for the taxonomic studies of genus Spathoglottis. My detailed comments are as follows:

1. Language

(1)The case of the word “Dwarf” should be uniformly case-sensitive in the text. For example, “dwarf” in line 496 should be capitalized.

(2) In line 350, the “cspecies” should be “species”.

(3) “Clade III-A” in line 497 should be “Clade III-B”.

Please check the manuscript carefully.

2. Structure

(1) Is it possible to combine some trees together as far as possible to avoid the confusion and fatigue by readers? There are so many figures in the MS.

3. Content

(1) The number of keywords can be reduced appropriately. Select and extract words from the paper that can show and reflect the theme of the full text. For example, you listed ITS and trnL-F, why not matK?

(2) Experimental method steps can be appropriately reduced, detailed slightly appropriate.

(3) Can the “yellow-flower species” mentioned in line 383 be marked in the picture , to show the two yellow-flower species (S.affinis_FAN 025 and S.kimballiana_FAN085) embedded within the Large Purple Spathoglottis group? Because it is mentioned in line 273 that “Taxa highlighted in yellow box are the yellow flower Spathoglottis while purple box comprises of flowers in different shades of purple”.

(4) Line 308-310: How did the authors define the size of plant and flower? I suggest that the authors should add the description of size. In addition, they should also illustrated which species belong to dwarf or large-sized, small, medium or large flower. The authors should label them after tip labels of each tree or a combined tree.

4. Article illustration

(1) First of all, the figures of Spathoglottis provided by the authors are very exquisite, which can well show the readers the morphological characteristics of Spathoglottis involved in this research. But the only drawback is that the layout of the figures is not very neat, the spacing between the pictures and position of the scale is not uniform.

(2) The figures of evolutionary tree provided clearly show the phylogenetic relationships of Spathoglottis involved in the study. It would be better if the text size, spacing and diagrams in the figures could be unified. Such as: in figures 5-10, the font size of “S. affinis FAN 025 (Thai-Myanmar Border)” is clearly inconsistent with the others. Figure 4: the branches are not the same thickness.

Author Response

Dear Reviewer,

Thank you so much for your constructive comments towards the betterment of our manuscript. We have made changes accordinglyas listed in the attachment:

Yours sincerely,

FAN, KS, RG, KNAM and ASO
